# Alternative Characterizations of Methyl Lucidone-Responsive Differentially Expressed Genes in *Drosophila melanogaster* Using DEG-by-Index Ratio Transformation

**DOI:** 10.3390/insects16090898

**Published:** 2025-08-27

**Authors:** Sang Woon Shin, Ji Ae Kim, Jun Hyoung Jeon, Kunhyang Park, SooJin Lee, Hyun-Woo Oh

**Affiliations:** 1Core Research Facility & Analysis Center, Korea Research Institute of Bioscience and Biotechnology, Daejeon 34141, Republic of Korea; jiaekim@kribb.re.kr (J.A.K.); kunhyang@kribb.re.kr (K.P.); 2Department of Microbiology and Molecular Biology, Chungnam National University, Daejeon 34134, Republic of Korea; leesoojin@cnu.ac.kr; 3Department of Southern Area Crop Science, National Institute of Crop Science, Rural Development Administration, Miryang 50424, Republic of Korea; wjs258@korea.kr

**Keywords:** methyl lucidone, RNA-Seq, normalization, DESeq2, edgeR

## Abstract

Standard normalization methods—such as relative log expression (RLE) via DESeq2 and trimmed mean of M-values (TMM) via edgeR—failed to consistently identify differentially expressed genes (DEGs) in our five primary RNA-Seq experiments and four validation datasets. This limitation prompted us to develop the DEG-by-index ratio transformation (DiRT), which is a novel normalization approach that significantly improves DEG detection and enables reliable validation across independent experiments.

## 1. Introduction

Methyl lucidone, a plant-derived diterpene from species like *Lindera erythrocarpa*, acts as a juvenile hormone (JH) disruptor in insects [1,2,3]. JHs are critical regulators of insect development, metamorphosis, and reproduction [4]. Disruption of JH signaling can lead to developmental arrest and mortality in insects [5].

RNA sequencing (RNA-Seq) has emerged as a powerful alternative to RT-qPCR for transcriptome-wide analysis [6], enabling comprehensive detection of differentially expressed genes (DEGs). However, RNA-Seq data exhibit significant technical and biological variations, necessitating normalization for reliable DEGs characterization. Widely used methods include DESeq2 (relative log expression, RLE) [7], edgeR (trimmed mean of M-values, TMM) [8], and limma/voom (precision weights derived from nonparametric mean–variance modeling) [9,10].

While traditional RNA-Seq workflows—typically involving three or more independent experiments with RLE/TMM normalization and RT-qPCR validation—have successfully identified DEGs in many studies, this approach failed in our investigation of methyl lucidone effects. Despite performing five independent RNA-Seq experiments using *Drosophila melanogaster* larvae, conventional methods only identified either two DEGs (DeSeq2) or none (edgeR), with no validation in four additional RNA-Seq datasets.

To address this limitation, we developed the DEG-by-index ratio transformation (DiRT): a novel method grounded in compositional data analysis (CDA) principles [11,12,13]. In DiRT, an index gene has an expression profile that is highly correlated with a target DEG under control (untreated) conditions, thereby serving as a stable internal reference unique to that specific DEG. To identify candidate index genes, we calculated the normalized standard deviation (NSD) of the expression ratio between each gene and every other gene across the control samples and ranked all ~10,000 genes accordingly. For each gene, the 10 candidates with the lowest NSD values—indicating the greatest similarity in expression under control conditions—were selected. We empirically observed that DEG–index gene pairings drawn from this low-NSD set consistently produced DiRT ratios that maximized discrimination between control and treatment groups. From the resulting ~100,000 (10 × 10,000) DEG–index gene combinations, the optimal pairing for each DEG was chosen based on the highest separation between groups. Unlike traditional single-reference normalization methods, which apply the same housekeeping or reference gene to all targets, the index gene in DiRT is selected individually for each DEG to preserve DEG-specific co-expression patterns. Crucially, the index gene must remain non-responsive to the experimental perturbation—in this case, methyl lucidone treatment—ensuring that changes in the DEG’s expression reflect true treatment effects rather than shared regulation or technical noise. This DEG-specific pairing provides a stable denominator for ratio-based normalization, thereby minimizing inter-sample variability and enhancing reproducibility across independent experiments.

We hypothesized that this gene-specific normalization strategy would enhance the robustness and reproducibility of DEG detection, particularly under conditions of high experimental variability, as supported by KEGG pathway analysis. Our results demonstrate that this approach can reveal reproducible, biologically relevant DEGs that remain undetected using conventional global normalization methods.

## 2. Materials and Methods

### 2.1. Drosophila melanogaster Sample Acquisition and Preparation for RNA-Seq

Twenty male and twenty female *Drosophila melanogaster* adults were placed in individual vials, each containing 3 g of artificial diet mixed with either 0.5% methyl lucidone (*w*/*v*) or 0.5% ethanol (*w*/*v*) as a control. After 2 days of oviposition, adult flies were removed, and eggs were allowed to develop. Second-instar larvae (4–5 days post-oviposition) were collected from each vial. Total RNA was isolated using the RNeasy isolation kit (Qiagen, Hilden, Germany) following the manufacturer’s protocols. RNA libraries were prepared for Illumina sequencing (San Diego, CA, USA).

### 2.2. Data Processing of 18 D. melanogaster RNA-Seqs

After sequencing, raw FASTQ files from all batches were processed through Galaxy server [14,15] pipelines. Single-end reads underwent quality control and adapter trimming with fastp (Galaxy Version 1.0.1+galaxy1) [16], followed by alignment to the dm6 genome [17] using HISAT2 (Galaxy Version 2.2.1+galaxy1) [18]. The mapped BAM files were then quantified with featureCounts (Galaxy Version 2.1.1+galaxy0) [19] to obtain the read count (RC) of each transcript (dm6 NCBI reference genes). General information on the 18 RNA-Seq datasets, including NCBI SRA accession numbers, is shown in Table 1.

### 2.3. DiRT Analysis Workflow

For the DiRT analysis, we selected the 10,000 most abundantly expressed genes from 17,868 annotated *D. melanogaster* (dm6) transcriptome genes based on their RCs across 10 RNA-Seq datasets (five control: C1–C5; five treated: T1–T5). This threshold minimized the inclusion of low-expression genes, which could produce unstable RC ratios and result in randomly selected or unreliable index genes.

For each target gene, the RC ratios were calculated by dividing its read count by that of every other gene in the dataset. From these ratios, we identified 10 gene pairs with the lowest normalized standard deviation (NSD) scores in the control samples as DEG–index or index–DEG pair candidates. This approach ensured the stability of the RC ratio under baseline conditions.

Using a custom Python 3 script (https://github.com/shinwoongg/DiRT-normalization), we generated a comprehensive DiRT candidate database comprising 100,000 columns. DiRT-normalized values (RC_target gene_/RC_another gene_) were used to compute *p*-values using a two-sample, two-tailed *t*-test, assuming equal variance between the control and treated groups.

To reduce false positives from DEG-DEG pairs, we excluded DiRTs with denominator genes that showed differential expressions (CPM-based *p*-value < 0.1). When multiple DiRT values were available for a single target gene, we retained only one entry with the lowest adjusted *p*-value (Benjamini–Hochberg correction, rank = 9988). This yielded a final dataset consisting of 9988 DiRT-normalized expression levels (see Appendix A).

### 2.4. DESeq2/edgeR Analyses

We used R version 4.3.1 together with the Bioconductor packages DESeq2 (v1.40.2) and edgeR (v3.42.4) for differential expression analyses. Parallel to the DiRT pipeline, we selected the same 10,000 most abundantly expressed genes from the *D. melanogaster* (dm6) transcriptome to ensure consistency across methods.

For the DESeq2 analysis, the raw RCs were normalized using the RLE method. Differential expression was assessed using the Wald test, and *p*-values were adjusted for multiple testing using the Benjamini–Hochberg false discovery rate (FDR) correction, as implemented within the DESeq2 framework (see Appendix A).

For the edgeR analysis, normalization was performed using the trimmed mean of M-values (TMM) method. Differential expressions were assessed using a two-sample, two-tailed *t*-test identical to the statistical approach applied in the DiRT analysis, assuming equal variance. *p*-values were adjusted using the Benjamini–Hochberg method, ranking 10,000 genes. Adjusted *p*-values greater than 1 were indicated as NA (see Appendix A).

### 2.5. Heatmaps

Heatmaps were generated to visualize the expression profiles of 50 genes selected by three different normalization methods: DiRT, DESeq2, and edgeR. For the top 50 DiRTs, numerator genes with the lowest adjusted *p*-values were used; for DESeq2, genes were selected based on the lowest *p*-values obtained from the Wald test; and for edgeR, genes with the lowest *p*-values from the *t*-test were chosen. Expression values were normalized relative to the mean expression across all RNA-Seq samples, and heatmaps were created using Python libraries including matplotlib, seaborn, and pandas. Red indicates expression values greater than 1 (above average) and blue represents values lower than 1 (below average).

### 2.6. Methods for p-Value Calculation and Adjustment

*p*-values were calculated using a two-tailed, two-sample *t*-test assuming equal variance; the control samples (C1–C5) were compared to treated samples (T1–T5) based on either TMM-normalized values (edgeR) or DiRT-normalized values. Multiple testing corrections were performed using the Benjamini–Hochberg procedure. The number of genes ranked for the adjustment was 9988 for DiRT. For the DESeq2 package, *p*-values and adjusted *p*-values were calculated using the internal Wald test. None of the adjusted *p*-values from the edgeR analysis and only two genes from the DESeq2 analysis reached significance at a threshold of 0.05; therefore, unadjusted *p*-values were used for subsequent analyses.

### 2.7. KEGG Pathway Analysis of DiRT-Derived DEGs

From the filtered DiRTs with validation-adjusted *p*-values below 0.05 (Appendix A), we manually identified the apparent DEGs. These DEGs were either consistently upregulated or downregulated across all nine independent experiments (Appendix A). The remaining putative DEGs in the DiRT set lacked complete directional concordance. KEGG pathway analysis was then performed separately for the upregulated DEGs and downregulated DEGs, using the Database for Annotation, Visualization, and Integrated Discovery (DAVID) [20].

## 3. Results

### 3.1. Analayes of Methyl-Lucidone-Treated D. melanogaster RNA-Seq Using DESeq2 and edgeR

We conducted RNA-Seq on second-instar *D. melanogaster* larvae treated with either methyl lucidone or ethanol (as a control) across five independent experiments. DESeq2 identified the following two DEGs at adjusted values of *p* < 0.05: *CG14265* and *Gyc76C* (Appendix A). EdgeR detected no significant DEGs under the same threshold (Appendix A).

Validation using four additional independent RNA-Seq datasets (four control and four treated datasets) showed no significant differential expressions for *CG14265* and *Gyc76C* (Figure 1a,b). Similarly, the top DEG candidate genes were identified using edgeR; the lowest *p*-values (*CG7414* and *CG13197*) exhibited no significance during validation (Figure 1c,d).

### 3.2. DEG-by-Index Ratio Transformation (DiRT) Normalization Identifies Reproducible DEGs

In contrast to the limited number of DEGs identified using DESeq2 and edgeR, DiRT normalization identified 1608 DiRTs with the ability to distinguish between control and treatment groups out of 9988 total DiRTs (Appendix A; adjusted *p* < 0.05). Heatmap analysis revealed clear separation between control and treated samples using DiRT-normalized values (Figure 2a); this pattern was not observed with DESeq2 (Figure 2b) or edgeR (Figure 2c). Moreover, the top 50 DiRT-derived DEGs—defined as numerator DEG–denominator index gene pairs—showed minimal overlap with the top candidates identified using DESeq2 or edgeR (Figure 2d).

Among the 1608 DiRTs, 280 met the validation criterion with adjusted *p*-values below 0.05 (Appendix A). DiRT normalization and further validation identified methyl lucidone-responsive DEGs that are undetectable using conventional methods. We identified two examples of DiRTs with the lowest adjusted *p*-values that also satisfied the validation criterion (*p* < 0.01) in Figure 3: (a) *CG9259*/*bond* and (b) *tim*/*MTF-1*. The validation of both DiRTs demonstrated complete consistency across the four control and treated samples (Figure 3).

The DEGs associated with these DiRTs were *CG9259* (Figure 4a) and *timeless* (*tim*) (Figure 4d), both of which showed consistent downregulation across all nine RNA-Seq datasets (five discovery and four validation datasets). In contrast, their respective index genes, *bond* (Figure 4b) and *MTF-1* (Figure 4e), displayed inconsistent or negligible differences in expression between control and treated samples. Control samples demonstrated strong correlations in expression between the DEG–index gene pairs, including the validated controls and discovery controls (e.g., *CG9259/bond*: Figure 4c; *tim/MTF-1*: Figure 4f).

Among the top 10 DiRTs with the lowest adjusted *p*-values meeting the validation criterion (*p* < 0.01), we identified eight DiRT-associated DEGs that exhibited consistent up- or downregulation across all nine independent RNA-Seq experiments (Appendix A; indicated in yellow). The respective index genes of these DiRTs displayed inconsistent or negligible differences in expression between control and treated samples (Appendix A). In these eight cases, the DiRT values of the DEG-index pairs were derived from either RC_DEG_/RC_index_ or RC_index_/RC_DEG_ calculations. The remaining two DiRTs consisted of DEG–DEG pairs: one showed consistent expression changes across all nine datasets (shown in green), and the other showed consistency in eight datasets (shown in orange). Notably, in both DEG–DEG pairs, the two genes were oppositely regulated.

## 4. Discussion

As this study’s primary goal was technical validation of DiRT, we did not pursue the deep biological interpretation of individual DEGs. Nevertheless, the DEGs identified via DiRT normalization exhibited distinct molecular characteristics.

From 280 validated DiRTs, we manually identified 109 strictly defined DEGs. These DEGs exhibited consistent regulation—86 entirely upregulated and 23 entirely downregulated genes—across all nine independent experiments (Appendix A). We conducted KEGG pathway analysis using DAVID, which revealed significant (adjusted *p* < 0.05) gene enrichment in drugs and xenobiotic metabolism via cytochrome P450 or other enzymes in the proteasome and methyl lucidone detoxification pathways (Table 2).

KEGG analysis of the 23 downregulated DEGs did not reveal significant pathway enrichment, likely due to the limited number of genes in this set. The number of downregulated genes—and thus the potential for detecting enriched pathways—could increase substantially if less stringent criteria for DEG definition were applied.

In insects, JH signaling is mediated by the Methoprene-tolerant bHLH-PAS receptor (Met) together with the coactivator Taiman (also known as SRC or FISC), and this complex activates JH-responsive genes such as *Kr-h1* [21]. In the mosquito *Aedes aegypti*, an MET/CYCLE (CYC) bHLH-PAS heterodimer links JH to circadian outputs in a light-dependent manner [22]. In honeybees (*Apis mellifera*), the juvenile hormone was shown to strengthen circadian rhythms and accelerate rhythm development in newly emerged worker bees [23]. In *Drosophila*, *tim* encodes a core clock protein that forms a complex with PERIOD (PER) to repress the transcriptional activators CLOCK–CYCLE (CLK–CYC) [24]. Our finding that *tim* is downregulated by the JH disruptor methyl lucidone suggests that JH signaling can intersect with the circadian clock through shared bHLH-PAS transcription factors and hormone-dependent modulation of clock gene expression, as seen in MET/CYC-mediated regulation in mosquitoes and hormone-induced changes in circadian outputs in other insects.

We observed that DEG–DEG pairs with opposite regulation could generate DiRT values with low adjusted *p*-values (Appendix A). Such combinations may represent artificial pairings that coincidentally satisfy the selection criteria. To minimize their occurrence, we applied a Wald test *p*-value threshold (>0.1) to the index gene; however, this approach did not fully eliminate DEG–DEG pairings. Nonetheless, only two of the top 10 DiRTs fell into this category, indicating that the majority of DiRTs are composed of genuine DEG–index or index–DEG pairs. This phenomenon likely occurs because dividing the expression of an upregulated gene by that of a downregulated gene (or vice versa) can amplify the treatment–control ratio, producing exceptionally small adjusted *p*-values even in the absence of a stable reference. While such artificial pairings are statistically possible, their low frequency in our dataset suggests minimal impact on overall DiRT performance. In many practical applications, the inclusion of a small number of these pairs is unlikely to affect the main biological conclusions, as they significantly contribute to the ranking of top candidates. Nevertheless, awareness of their potential presence remains important for accurate interpretation of DiRT results.

An additional consideration is pleiotropy, where a single DEG may influence multiple biological processes or pathways simultaneously [25]. In the context of methyl lucidone–responsive DEGs, such pleiotropic effects could mean that the observed transcriptional changes are driven by overlapping roles in different pathways, making direct functional attributions more challenging. Although this complexity does not alter the statistical robustness of DiRT, it highlights the need for careful biological interpretation, particularly for DEGs with known multifunctional roles.

The differences between Figure 1 and Figure 4 reflect the effect of normalizing with a different number of RNA-Seq datasets—10 discovery datasets in the former versus all 18 datasets in the latter. For conventional normalization methods such as RLE and TMM, scaling factors are estimated from the entire set of samples under analysis. The addition of new datasets can alter these scaling factors, particularly if the added samples differ in sequencing depth, library composition, or batch-specific biases, leading to subtle shifts in normalized values for all samples. In contrast, DiRT normalization is based on the DEG–index gene ratios computed within each dataset, which reduces the influence of unrelated samples and minimizes the propagation of batch effects from additional datasets. Although both approaches are influenced by sample composition, DiRT remained relatively stable across dataset expansions. This highlights its potential advantage in multi-batch RNA-Seq studies.

DiRT normalization identified numerous DEGs with successful validation, demonstrating its capability to detect methyl lucidone-responsive genes. As the primary focus of this study was the technical validation of DiRT, we did not pursue an in-depth biological interpretation of individual DEGs. Nevertheless, DiRT normalization identified 1608 putative DEGs across five discovery RNA-Seq experiments; 287 of these DiRTs were validated using four independent validation datasets. We expect that a substantial portion of these 287 DiRTs represent genuine DEGs, given that DEGs from 8 of the top 10 DiRTs showed consistent regulation across all nine RNA-Seq experiments. The potential biological relevance of these DEGs to methyl lucidone treatment remains to be explored in future studies.

The partial or failed validation observed in some DiRTs likely stems from the limited scope of initial discovery experiments; this may prove incapable of fully capturing biological and technical variability. Nevertheless, DiRT yielded markedly higher DEG discovery rates compared to conventional methods. Unlike global normalization approaches, DiRT’s gene-specific strategy effectively mitigated batch effects in cases where DESeq2 and edgeR failed.

## 5. Conclusions

In conclusion, we developed the DiRT method, a gene-specific normalization approach that identifies robust DEGs in *D. melanogaster* RNA-seq datasets under various experimental conditions. By leveraging pairwise normalization with dynamically selected index references, DiRT effectively minimizes background noise and significantly improves the reproducibility of DEG detection across datasets. Compared to conventional normalization methods such as DESeq2 and edgeR, DiRT identified distinct sets of DEGs with validation and clearer expression patterns, demonstrating its applicability to complex systems where experimental heterogeneity obscures transcriptional responses. To further establish its generality, DiRT should be tested using RNA-seq datasets from additional species and diverse experimental designs. Moreover, functional studies—such as targeted knockdowns of top-ranked DiRT-identified DEGs using RNA interference or CRISPR, followed by phenotypic analysis or biochemical assays—could provide direct evidence linking transcriptional changes to functional outcomes.

## Figures and Tables

**Figure 1 insects-16-00898-f001:**
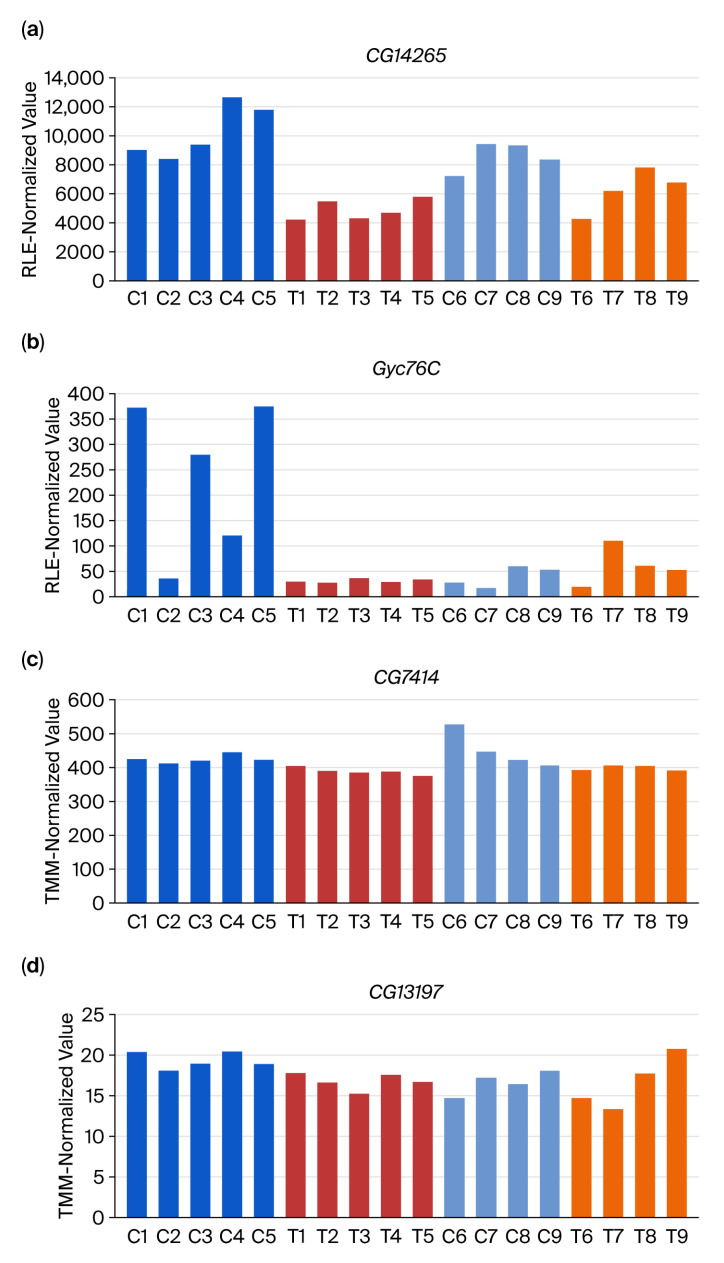
Validation of top DEG candidates identified using DESeq2 and edgeR. Expression profiles of the two most significant DEGs from (**a**,**b**) DESeq2 (CG14265, Gyc76C) and (**c**,**d**) edgeR (CG7414, CG13197) analyses. Initial analysis used 10 RNA-Seq datasets (C1–C5 controls, T1–T5 treated with methyl lucidone; blue/red bars). Validation in 8 independent datasets (C6–C9 controls, T6–T9 treated; light blue/orange bars). RLE: relative log expression (used in DESeq2); TMM: trimmed mean of M-values (used in edgeR); DEG: differentially expressed gene.

**Figure 2 insects-16-00898-f002:**
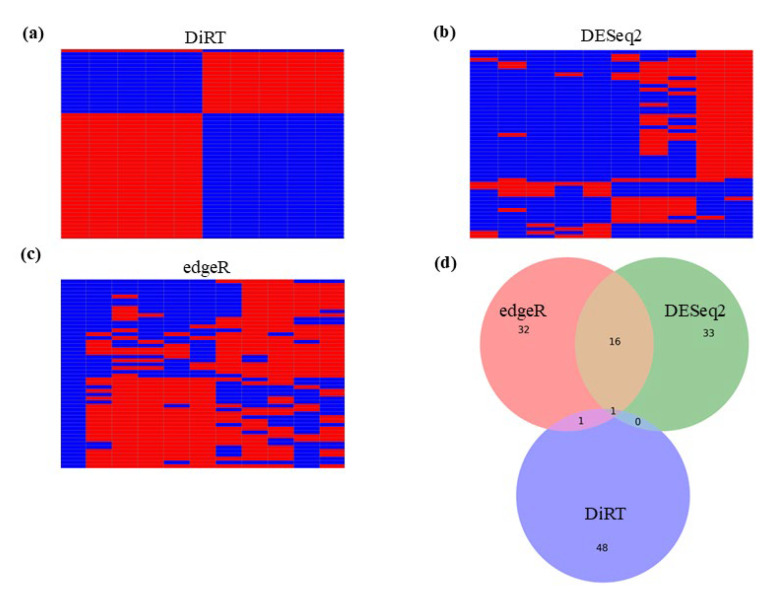
Heatmaps of the top 50 measures identified using DiRT, DESeq2, and edgeR, showing minimal overlap between DiRT-derived DEGs and the genes with the lowest *p*-values obtained using DESeq2/edgeR (**a**) Heatmap of the top 50 DiRT-normalized measures ranked based on the lowest *p*-values. (**b**) Heatmap of the top 50 genes identified using DESeq2 based on the lowest *p*-values. (**c**) Heatmap of the top 50 genes identified using edgeR based on the lowest *p*-values. (**d**) Venn diagram illustrating the top 50 DiRT-derived DEGs with the lowest *p*-values showing minimal overlap. The top 50 genes were identified using DESeq2 or edgeR based on the lowest *p*-values. All measures in (**a**–**c**) were normalized to the average values across 10 RNA-Seq samples (C1–C5 for controls; T1–T5 for treated samples). Red indicates values greater than 1, and blue indicates values less than 1. DEG: differentially expressed gene; DiRT: DEG-by-index ratio transformation.

**Figure 3 insects-16-00898-f003:**
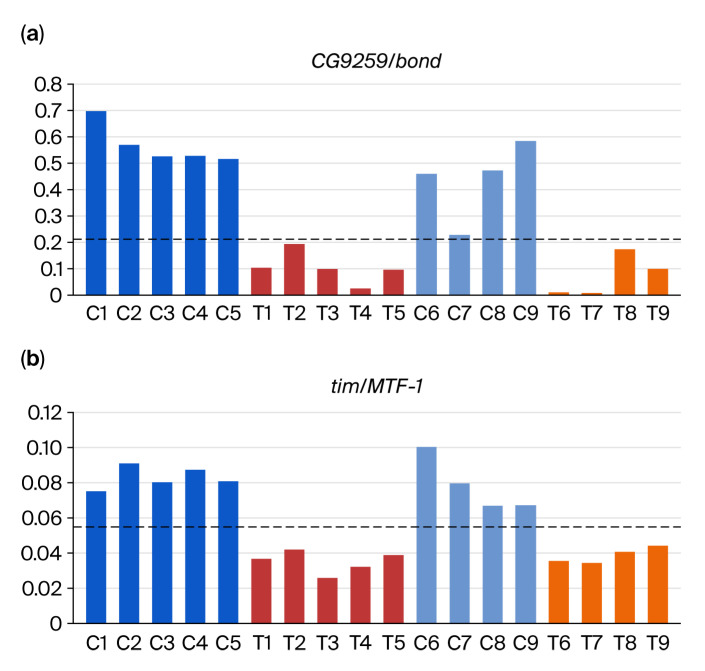
Discovery and validation of DiRTs. DiRT values—calculated as the ratio of read counts between a putative DEG and its corresponding index gene—were determined using five initial RNA-Seq experiments and subsequently validated in four independent RNA-Seq datasets. Both panels (**a**,**b**) illustrate cases of complete validation, with dotted lines representing the predefined threshold for validation. The Y-axis denotes DiRT values. DEG: differentially expressed gene; DiRT: DEG-by-index ratio transformation.

**Figure 4 insects-16-00898-f004:**
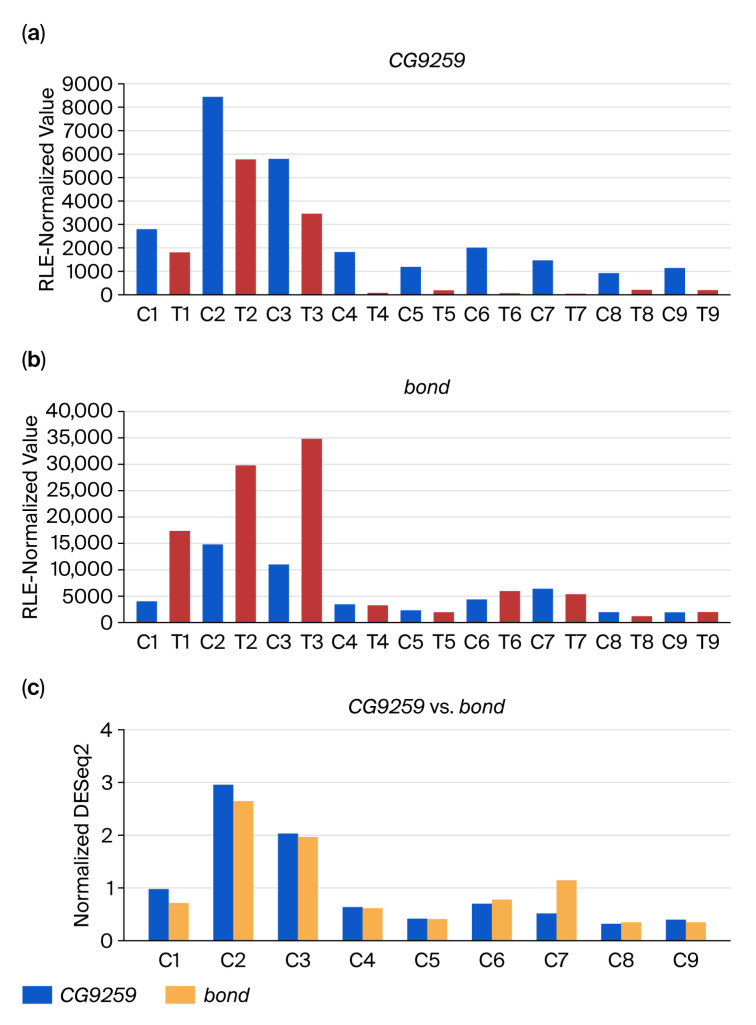
Expression patterns of DiRT-identified DEGs and their corresponding index genes. Expression profiles of two DiRT-associated DEGs and their corresponding index genes, selected as the top two DiRTs with the lowest adjusted *p*-values from the initial screening performed across 10 RNA-Seq datasets; they were validated with a *p*-value threshold of <0.01 using eight additional RNA-Seq datasets. Panels (**a**) and (**d**) show the RLE-normalized expression of DEGs *CG9259* and *tim*, respectively, while panels (**b**,**e**) show the RLE-normalized expression of their corresponding index genes *bond* and *MTF-1*. Blue bars represent control samples from both the discovery set (C1–C5) and the validation set (C6–C9); red bars represent methyl lucidone-treated samples from the discovery set (T1–T5) and validation set (T6–T9). Panels (**c**,**f**) illustrate strong expression correlations between DEG–index gene pairs across all nine control samples, evident in both discovery and validation controls. To evaluate these correlations (y-axis: normalized DESeq2), RLE-normalized expression values were further scaled relative to the average RLE values computed across nine independent control RNA-Seq datasets. RLE: relative log expression (used in DESeq2); DEG: differentially expressed gene; DiRT: DEG-by-index ratio transformation.

**Table 1 insects-16-00898-t001:** General information and SRA submission numbers of 18 *Drosophila melanogaster* RNA-seq data.

Sample ID	Reads Assigned to *dm6*-Annotated Genes	NCBI Accession #	Condition
C1	67,399,437	SRR22894279	EtOH-treated
C2	94,044,531	SRR22891343	EtOH-treated
C3	92,115,653	SRR22891368	EtOH-treated
C4	68,365,272	SRR22891367	EtOH-treated
C5	74,929,527	SRR22891357	EtOH-treated
T1	50,864,597	SRR22891355	Methyl lucidone-treated
T2	91,480,523	SRR22891354	Methyl lucidone-treated
T3	92,471,264	SRR22891353	Methyl lucidone-treated
T4	67,774,651	SRR22891366	Methyl lucidone-treated
T5	76,983,129	SRR22891362	Methyl lucidone-treated
C6	65,937,883	SRR22891360	EtOH-treated
C7	62,985,695	SRR22891359	EtOH-treated
C8	63,522,838	SRR22891358	EtOH-treated
C9	69,304,669	SRR22891356	EtOH-treated
T6	66,503,607	SRR22891364	Methyl lucidone-treated
T7	66,182,329	SRR22891363	Methyl lucidone-treated
T8	69,941,736	SRR22891361	Methyl lucidone-treated
T9	65,816,863	SRR22891365	Methyl lucidone-treated

**Table 2 insects-16-00898-t002:** KEGG pathway analysis in 86 DiRT-derived downregulated DEGs.

KEGG Pathways	Gene Count	*p*-Value	Benjamini
Proteasome	8	2.8 × 10^−6^	0.00015
Drug metabolism—cytochrome P450	8	0.000015	0.00033
Metabolism of xenobiotics via cytochrome P450	8	0.000019	0.00033
Drug metabolism—other enzymes	7	0.001	0.014

## Data Availability

The RNA-Seq datasets were deposited in the NCBI SRA database. The accession numbers are listed in Table 1. The Python script used to generate the DiRT candidate databases is available in the GitHub repository [https://github.com/shinwoongg/SHIN].

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
