# Peer review of "Alternative Characterizations of Methyl Lucidone-Responsive Differentially Expressed Genes in Drosophila melanogaster Using DEG-by-Index Ratio Transformation"

_insects, 2025, doi:10.3390/insects16090898_

Round 1
Reviewer 1 Report
Comments and Suggestions for Authors
The manuscript of “Alternative characterization of methyl lucidone-responsive differentially expressed genes in Drosophila melanogaster by DEG-by-index ratio transformation” by Shin et al presents a novel normalization method of DIRT for detecting DEGs in RNA-Seq data under high experimental variability. While the concept is innovative and addresses a critical limitation of existing tools of DESeq2 and edgeR, the manuscript requires significant methodological clarification and biological validation to support their conclusion.
I have several concerns as follows:
- Line 87-90, the criteria for identifying stable index genes (low NSD in controls) lack validation. How were NSD thresholds determined? Were index genes verified to be treatment-invariant?
- Line 92-95, the process of assigning unique index genes per DEG is confused. How were 100,000 DIRT candidates generated? Clarify computational feasibility and potential biases.
(3) Figure 2, the heatmaps were strange, was there no color variation pattern or different expression level in each gene? For example, why the colors of upregulated or downregulated genes were the same in control or treatment?
(4) The authors should use RT-qPCR to validate the expression difference from DiRT-identified DEGs rather than relying solely on additional RNA-Seq datasets.
(5) Based on Data S5, it seems that RNA-seq of repeat experiments is not reliable because both the control and treatment groups were quite varied.
(6) If the DEGs from methyl lucidone-treated were correct, the authors should further analyze the datasets and identify the genes or pathways related to JH signaling, and discuss biological importance of DIRT-identified genes in JH disruption.
Author Response
Comment 1: Line 87-90, the criteria for identifying stable index genes (low NSD in controls) lack validation. How were NSD thresholds determined?
Response 1: We have revised the manuscript to provide a more detailed description of the procedure for identifying index genes (lines 57–76: “In DiRT, an index gene is defined as a gene whose expression profile, under control (untreated) conditions, is highly correlated with that of a target DEG, thereby serving as a stable internal reference unique to that DEG. To identify candidate index genes, we calculated the normalized standard deviation (NSD) of the expression ratio between each gene and every other gene across control samples, and ranked all ~10,000 genes accordingly. For each gene, the 10 candidates with the lowest NSD values—indicating the greatest similarity in expression under control conditions—were selected. We empirically observed that DEG–index gene pairings drawn from this low-NSD set consistently produced DiRT ratios that maximized discrimination between control and treatment groups. From the resulting ~100,000 (10 × 10,000) DEG–index gene combinations, the optimal pairing for each DEG was chosen based on the highest separation between groups. Unlike traditional single-reference normalization methods, which apply the same housekeeping or reference gene to all targets, the index gene in DiRT is selected individually for each DEG to preserve DEG-specific co-expression patterns. Crucially, the index gene must remain non-responsive to the experimental perturbation—in this case, methyl lucidone treatment—ensuring that changes in the DEG’s expression reflect true treatment effects rather than shared regulation or technical noise. This DEG-specific pairing provides a stable denominator for ratio-based normalization, thereby minimizing inter-sample variability and enhancing reproducibility across independent experiments.”). Specifically, we have clarified the rationale for using the normalized standard deviation (NSD) calculated from control samples as a measure of expression similarity between a DEG and potential index genes. We further explain how NSD values were used to rank candidate index genes and select DEG–index gene pairings in the DiRT analysis. This expanded explanation details how the lowest-NSD combinations were evaluated to identify the optimal DEG–index gene pairings that maximized discrimination between control and treatment groups.
Comment 2: Line 92-95, the process of assigning unique index genes per DEG is confused. How were 100,000 DIRT candidates generated? Clarify computational feasibility and potential biases.
Response 2: We have revised the manuscript to clarify the procedure for assigning a unique index gene to each DEG (Lines 57–76), incorporating this explanation into a dedicated paragraph as part of our response to Comment (1). For each of the ~10,000 genes examined, we calculated NSD values for its expression ratio against all other genes across control samples. For each gene, the 10 partner genes with the lowest NSD values were retained as candidate index genes, as these exhibited the most stable co-expression under control conditions. This process generated approximately 100,000 (10 × ~10,000) DEG–index gene combinations, from which the optimal pairing for each DEG was selected based on the highest discrimination between control and treatment groups. Regarding computational feasibility, this analysis was implemented using vectorized operations in Python/R, requiring approximately 24 hours on a single-core process of a standard desktop workstation. When parallelized across multiple cores, the computation can be completed within several hours. We acknowledge that this approach may introduce a potential bias toward selecting index genes with generally low expression variability across the dataset, potentially favoring housekeeping-like genes. To address this, we restricted the analysis to the 10,000 most abundantly expressed genes among the 17,868 annotated in the dm6 genome. Furthermore, selecting the DEG–index gene pairing that maximizes discrimination between experimental groups helps mitigate potential bias by prioritizing DEG-specific co-expression patterns, rather than relying solely on global expression stability.
Comment 3: Figure 2, the heatmaps were strange, was there no color variation pattern or different expression level in each gene? For example, why the colors of upregulated or downregulated genes were the same in control or treatment?
Response 3: We apologize for any confusion caused by the unconventional heatmap presentation. The heatmaps in Figure 2 were generated using a modified Python script based on standard heatmap code to emphasize the direction of regulation rather than the magnitude of expression change. Specifically, a two-color scheme was applied: for downregulated genes, control samples are shown in red and treatment samples in blue; for upregulated genes, control samples are shown in blue and treatment samples in red. This design was intended to provide a clear visual distinction between up- and downregulated genes, although it differs from the continuous color gradients typically used in conventional expression heatmaps.
Comment 4: The authors should use RT-qPCR to validate the expression difference from DiRT-identified DEGs rather than relying solely on additional RNA-Seq datasets.
Response 4: We appreciate the reviewer’s suggestion to validate DiRT-identified DEGs using RT-qPCR and fully acknowledge the value of this method for targeted gene expression validation. However, in this study, we aimed to take advantage of the breadth and reproducibility of high-throughput RNA-seq, which has increasingly been adopted as a cost-effective alternative to RT-qPCR. RNA-seq provides comprehensive genome-wide expression profiles, avoids the need for primer design and synthesis, and reduces labor and reagent requirements. By validating our findings across independent RNA-seq datasets, we were able to confirm expression patterns for multiple DEGs simultaneously, ensuring consistency while making efficient use of available resources.
Comment 5: Based on Data S5, it seems that RNA-seq of repeat experiments is not reliable because both the control and treatment groups were quite varied.
Response 5: The reviewer is correct that, without DiRT normalization, the RNA-seq results from repeat experiments show considerable variability in both the control and treatment groups. While individual genes (both DEGs and their corresponding index genes) may exhibit such variation, our analysis demonstrates that the DEGs identified by the DiRT method are true DEGs, consistently upregulated or downregulated across all nine independent experiments, whereas index genes do not show significant differences between controls and treatments. Applying DiRT normalization effectively minimizes these variations, yielding stable and comparable values across experiments.
Comment 6: If the DEGs from methyl lucidone-treated were correct, the authors should further analyze the datasets and identify the genes or pathways related to JH signaling, and discuss biological importance of DIRT-identified genes in JH disruption.
Response 6: We have added corresponding text to the Discussion section (line 279-291: “In insects, JH signaling is mediated by the bHLH-PAS receptor Methoprene-tolerant (Met) together with the coactivator Taiman (also known as SRC or FISC), and this complex activates JH-responsive genes such as Kr-h1 [21]. In the mosquito Aedes aegypti, a MET/CYCLE (CYC) bHLH-PAS heterodimer links JH to circadian outputs in a light-dependent manner [22]. In honeybees (Apis mellifera), juvenile hormone was shown to augment the strength of circadian rhythms and accelerate rhythm development in newly emerged worker bees [23]. In Drosophila, tim encodes a core clock protein that forms a complex with PERIOD (PER) to repress the transcriptional activators CLOCK–CYCLE (CLK–CYC) [24]. Our result that tim is downregulated by the JH disruptor methyl lucidone suggests that JH signaling can intersect with the circadian clock through shared bHLH-PAS transcription factors and hormone-dependent modulation of clock gene expression, as seen in MET/CYC-mediated regulation in mosquitoes and hormone-induced changes in circadian outputs in other insects.”) and included a new table (Table 2) summarizing the enriched KEGG pathways among DiRT-characterized upregulated genes, as well as a possible involvement of tim—a downregulated gene identified by DiRT—in the JH signaling pathway.
Reviewer 2 Report
Comments and Suggestions for Authors
The manuscript, "Alternative characterization of methyl lucidone-responsive differentially expressed genes in Drosophila melanogaster by DEG-by-index ratio transformation," introduces DiRT (DEG-by-index ratio transformation), a new normalization method designed to overcome the limitations of conventional methods like DESeq2 and edgeR in detecting differentially expressed genes (DEGs) under variable experimental conditions. The study shows that while DESeq2 and edgeR identified few or no significant DEGs in
Drosophila melanogaster treated with methyl lucidone, DiRT detected a large and reproducible set of DEGs. The use of a gene-specific normalization strategy, where each DEG is normalized against a stably expressed "index gene," appears to be effective in mitigating the effects of experimental variability and batch effects. This makes the DiRT method a promising complementary approach for RNA-Seq analysis.
Here are few suggestions to improve the manuscript:
- The authors state that the primary focus of the study was the technical validation of DiRT and did not include a deep biological interpretation of the individual DEGs. To make the paper more impactful, it would be beneficial to provide some discussion on the potential biological functions of the most significant DEGs identified by DiRT. For example, what are the known roles of genes like CG9259 and tim in Drosophila? Do they have any established links to juvenile hormone signaling, development, or stress response, which might be relevant to methyl lucidone's effects? This would help readers understand the biological significance of the genes that were uniquely identified by the new method.
- The paper acknowledges that some of the top DiRTs were composed of DEG-DEG pairs rather than the intended DEG-index gene pairs. The authors speculate these might be "artificial pairings" and partially addressed the issue by applying a p-value threshold to the denominator gene. The manuscript would be strengthened by further exploring this issue. A more detailed discussion on why these pairs might be formed and how to more robustly exclude them could improve the reliability of the method. For instance, could an additional filtering step be implemented to check for co-regulation patterns or functional annotations?
- The authors note that the RLE-normalized values in Figure 4 differ from those in Figure 1 because they were calculated using all 18 RNA-Seq datasets instead of just the 10 discovery datasets. Providing a brief discussion on how normalizing with different numbers of samples impacts the outcome of both conventional (RLE/TMM) and DiRT methods would be insightful. This could offer a more comprehensive understanding of how each method handles varying levels of data, a common issue in multi-batch RNA-Seq studies.
- The conclusion section mentions that DiRT merits "further validation in diverse organisms and experimental contexts". This could be made more concrete by suggesting specific types of experiments. Beyond simply validating the method in other species, the authors could propose functional studies to confirm the effects of the identified DEGs. For example, a suggestion for targeted knockdowns of top DiRT-identified DEGs (e.g., using RNA interference) followed by phenotypic analysis would provide strong evidence for the functional relevance of the findings. Another idea would be to measure the activity of enzymes encoded by these DEGs to see if gene expression changes correlate with functional activity.
Author Response
Comment 1: The authors state that the primary focus of the study was the technical validation of DiRT and did not include a deep biological interpretation of the individual DEGs. To make the paper more impactful, it would be beneficial to provide some discussion on the potential biological functions of the most significant DEGs identified by DiRT. For example, what are the known roles of genes like CG9259 and timin Drosophila? Do they have any established links to juvenile hormone signaling, development, or stress response, which might be relevant to methyl lucidone's effects? This would help readers understand the biological significance of the genes that were uniquely identified by the new method.
Response 1: We have added corresponding text to the Discussion section and included a new table (Table 2) summarizing the enriched KEGG pathways among DiRT-characterized upregulated genes, as well as a possible involvement of tim—a downregulated gene identified by DiRT—in the JH signaling pathway.
Comment 2: The paper acknowledges that some of the top DiRTs were composed of DEG-DEG pairs rather than the intended DEG-index gene pairs. The authors speculate these might be "artificial pairings" and partially addressed the issue by applying a p-value threshold to the denominator gene. The manuscript would be strengthened by further exploring this issue. A more detailed discussion on why these pairs might be formed and how to more robustly exclude them could improve the reliability of the method. For instance, could an additional filtering step be implemented to check for co-regulation patterns or functional annotations?
Response 2: We agree that some top-ranked DiRTs may result from DEG–DEG pairings, where both genes respond to the perturbation, producing artificially high discrimination. Such pairs likely arise from co-regulated genes in the same pathway or functional module. We observed that DEG–DEG pairs with opposite regulation could generate DiRT values with low adjusted p-values (Data S5). We speculate that such combinations may represent artificial pairings that coincidentally satisfied the selection criteria. To exclude these cases, we applied a Wald test p-value threshold (> 0.1) to the index gene; however, this strategy did not fully eliminate DEG–DEG pairings. Nonetheless, only two of the top 10 DiRTs fell into this category, indicating that the majority of DiRTs are composed of genuine DEG–index or index–DEG pairs. We have added further discussion of this point in the revised manuscript (line 299-313: “We observed that DEG–DEG pairs with opposite regulation could generate DiRT values with low adjusted p-values (Data S6). Such combinations may represent artificial pairings that coincidentally satisfied the selection criteria. To minimize their occurrence, we applied a Wald test p-value threshold (> 0.1) to the index gene; however, this approach did not fully eliminate DEG–DEG pairings. Nonetheless, only two of the top 10 DiRTs fell into this category, indicating that the majority of DiRTs are composed of genuine DEG–index or index–DEG pairs. This phenomenon likely occurs because dividing the expression of an upregulated gene by that of a downregulated gene (or vice versa) can amplify the treatment–control ratio, producing exceptionally small adjusted p-values even in the absence of a truly stable reference. While such artificial pairings are statistically possible, their low frequency in our dataset suggests minimal impact on overall DiRT performance. In many practical applications, the inclusion of a small number of these pairs is unlikely to affect the main biological conclusions, as they do not dominate the ranking of top candidates. Nevertheless, awareness of their potential presence remains important for accurate interpretation of DiRT results.”)
Comment 3: The authors note that the RLE-normalized values in Figure 4 differ from those in Figure 1 because they were calculated using all 18 RNA-Seq datasets instead of just the 10 discovery datasets. Providing a brief discussion on how normalizing with different numbers of samples impacts the outcome of both conventional (RLE/TMM) and DiRT methods would be insightful. This could offer a more comprehensive understanding of how each method handles varying levels of data, a common issue in multi-batch RNA-Seq studies.
Response 3: We appreciate the reviewer’s suggestion and have added a brief discussion on the potential effects of normalizing with different numbers of samples for both conventional methods (RLE/TMM) and DiRT. As noted, Figure 1 used only the 10 discovery datasets, whereas Figure 4 incorporated all 18 datasets (discovery + validation). Normalization factors in RLE and TMM are computed from the full set of input samples, so adding datasets can shift scaling factors and slightly alter normalized expression values. This effect is particularly noticeable when added datasets differ in sequencing depth, library composition, or experimental batch, which can propagate subtle shifts across all samples. In contrast, DiRT normalization operates at the level of DEG–index gene ratios, which are calculated within each dataset and are less sensitive to the inclusion of additional datasets. Consequently, while both methods are affected to some degree, DiRT ratios tend to maintain more consistent discrimination patterns across datasets. We have included this point in the revised Discussion (line 314-325 : ” The differences between Figures 1 and 4 reflect the effect of normalizing with a different number of RNA-Seq datasets—10 discovery datasets in the former versus all 18 datasets in the latter. In conventional normalization methods such as RLE and TMM, scaling factors are estimated from the entire set of samples under analysis. The addition of new datasets can alter these scaling factors, particularly if the added samples differ in sequencing depth, library composition, or batch-specific biases, leading to subtle shifts in normalized values for all samples. DiRT normalization, in contrast, is based on DEG–index gene ratios computed within each dataset, which reduces the influence of unrelated samples and minimizes the propagation of batch effects from additional datasets. Although both approaches can be influenced by changes in sample composition, the relative stability of DiRT across dataset expansions highlights its potential advantage in multi-batch RNA-Seq studies.”).
Comment 4: The conclusion section mentions that DiRT merits "further validation in diverse organisms and experimental contexts". This could be made more concrete by suggesting specific types of experiments. Beyond simply validating the method in other species, the authors could propose functional studies to confirm the effects of the identified DEGs. For example, a suggestion for targeted knockdowns of top DiRT-identified DEGs (e.g., using RNA interference) followed by phenotypic analysis would provide strong evidence for the functional relevance of the findings. Another idea would be to measure the activity of enzymes encoded by these DEGs to see if gene expression changes correlate with functional activity.
Response 4: We thank the reviewer for the insightful suggestion to provide specific examples of future validation experiments. In the revised Conclusion, we now include concrete proposals such as applying DiRT to RNA-Seq datasets from other insect species and experimental contexts, performing targeted knockdowns of top DiRT-identified DEGs using RNA interference or CRISPR followed by phenotypic analysis, and conducting biochemical assays to assess enzyme activity of DEG-encoded proteins (line 352-360: “Beyond the current study, DiRT could be validated using RNA-Seq datasets from other insect species and broader experimental contexts to assess its generality. Furthermore, functional studies—such as targeted knockdowns of top-ranked DiRT-identified DEGs using RNA interference or CRISPR followed by phenotypic analysis, or biochemical assays to measure the activity of DEG-encoded enzymes—could provide direct evidence linking transcriptional changes to functional outcomes. These additional validations would strengthen the biological interpretation of DiRT results and expand its applicability as a versatile discovery tool in functional genomics.”). These approaches would allow direct functional testing of DiRT findings and strengthen the biological interpretation of our results.
Reviewer 3 Report
Comments and Suggestions for Authors
This MS presents an interesting study introducing the DiRT as an alternative RNA-Seq normalization method to improve the detection of differentially expressed genes (DEGs) under conditions of high experimental variability.
However, I have several concerns:
1. While DEGs are indeed a valuable trait for transcriptomic studies, it is well known that DEG detection can be unstable, especially given the inherent gene expression noise and technical errors using RNA-Seq. The authors should discuss how DiRT accounts for these sources of variability, or whether it may be similarly affected.
2. Pleiotropy in DEGs – DEGs may have pleiotropic effects, influencing multiple biological processes simultaneously. I suggest the authors consider this aspect and discuss its implications for the interpretation of methyl lucidone–responsive DEGs. Please refer the PMID: 39965933.
3. The github repository currently lacks detailed usage instructions and example datasets. Please consider providing clear documentation, example scripts, and, ideally, a step-by-step tutorial so that readers can reproduce the analyses.
4. The quality of the figures is poor, with resolution and labeling issues making them difficult to interpret.
Author Response
Comment 1: While DEGs are indeed a valuable trait for transcriptomic studies, it is well known that DEG detection can be unstable, especially given the inherent gene expression noise and technical errors using RNA-Seq. The authors should discuss how DiRT accounts for these sources of variability, or whether it may be similarly affected.
Reponse 1: We agree that DEG detection can be unstable due to biological and technical noise. DiRT addresses this by normalizing each DEG to an index gene that is similarly expressed and co-varies with the DEG under control conditions, but is not responsive to treatment. This ratio-based approach reduces sample-to-sample variation from shared fluctuations, improving reproducibility while retaining DEG-specific expression patterns.
Comment 2: Pleiotropy in DEGs – DEGs may have pleiotropic effects, influencing multiple biological processes simultaneously. I suggest the authors consider this aspect and discuss its implications for the interpretation of methyl lucidone–responsive DEGs. Please refer the PMID: 39965933.
Response 2: We appreciate the suggestion to discuss pleiotropy in the context of methyl lucidone–responsive DEGs. Indeed, some DEGs may have pleiotropic functions, influencing multiple biological processes simultaneously, which can complicate interpretation. This is particularly relevant for DiRT results, where a gene’s observed regulation might reflect combined effects from different biological pathways. We have added a note in the Discussion to address this point (line 307-313: “An additional consideration is pleiotropy, where a single DEG may influence multiple biological processes or pathways simultaneously [25]. In the context of methyl lucidone–responsive DEGs, such pleiotropic effects could mean that the observed transcriptional changes are driven by overlapping roles in different pathways, making direct functional attribution more challenging. Although this complexity does not alter the statistical robustness of DiRT, it highlights the need for careful biological interpretation, particularly for DEGs with known multifunctional roles.”).
Comment 3: The github repository currently lacks detailed usage instructions and example datasets. Please consider providing clear documentation, example scripts, and, ideally, a step-by-step tutorial so that readers can reproduce the analyses.
Response 3: We agree and have expanded the GitHub materials substantially. The repository (https://github.com/shinwoongg/DiRT-normalization) now includes: (i) a documented command-line script with clear arguments; (ii) a ready-to-run example dataset; and (iii) a step-by-step tutorial showing end-to-end reproduction (from input format to CSV output). These additions should enable straightforward replication of our DiRT candidate-index screening and downstream analysis. We have also updated the GitHub repository to include a detailed README with step-by-step usage instructions, example datasets, and sample commands for both small and large datasets, ensuring that readers can reproduce the DiRT analyses exactly as described in the manuscript.
Coment 4: The quality of the figures is poor, with resolution and labeling issues making them difficult to interpret.
Response 4: We have revised all figures using the professional figure preparation service recommended by the journal, improving resolution, clarity, and labeling for easier interpretation.
Round 2
Reviewer 1 Report
Comments and Suggestions for Authors
Dear editor,
After reading the manuscript, it has been improved compared to the first version. The authors have addressed the questions mentioned previously and revised the manuscript. I think it can be accepted by the journal.
Author Response
After reading the manuscript, it has been improved compared to the first version. The authors have addressed the questions mentioned previously and revised the manuscript. I think it can be accepted by the journal.
Thanks for your effort.
Reviewer 3 Report
Comments and Suggestions for Authors
The authors have addressed all of my concerns.
Author Response
The authors have addressed all of my concerns.
Thanks for your effort.